# The Arabidopsis Hypoxia Inducible *AtR8* Long Non-Coding RNA also Contributes to Plant Defense and Root Elongation Coordinating with WRKY Genes under Low Levels of Salicylic Acid

**DOI:** 10.3390/ncrna6010008

**Published:** 2020-02-26

**Authors:** Shuang Li, Saraswati Nayar, HuiYuan Jia, Sanjay Kapoor, Juan Wu, Yasushi Yukawa

**Affiliations:** 1Graduate School of Science, Nagoya City University, Nagoya 467-8501, Japan; s.ri@nsc.nagoya-cu.ac.jp (S.L.); jia@nsc.nagoya-cu.ac.jp (H.J.); 2Department of Plant Molecular Biology, University of Delhi South Campus, Benito Juarez Road, New Delhi 110021, India; saraswatin@rgcb.res.in (S.N.); kapoors@genomeindia.org (S.K.); 3Key Laboratory of Saline-Alkali Vegetation Ecology Restoration, Ministry of Education, College of Life Science, Northeast Forestry University, Harbin 150040, China; wuj1970@163.com

**Keywords:** Arabidopsis, *AtR8* lncRNA, defense, long non-coding RNA, NPR1, PR-1, *Pseudomonas syringae*, root elongation, salicylic acid, WRKY

## Abstract

*AtR8* lncRNA was previously identified in the flowering plant *Arabidopsis thaliana* as an abundant Pol III-transcribed long non-coding RNA (lncRNA) of approximately 260 nt. *AtR8* lncRNA accumulation is responsive to hypoxic stress and salicylic acid (SA) treatment in roots, but its function has not yet been identified. In this study, microarray analysis of an *atr8* mutant and wild-type Arabidopsis indicated a strong association of *AtR8* lncRNA with the defense response. *AtR8* accumulation exhibited an inverse correlation with an accumulation of two *WRKY* genes (*WRKY53*/*WRKY70*) when plants were exposed to exogenous low SA concentrations (20 µM), infected with *Pseudomonas syringae*, or in the early stage of development. The highest *AtR8* accumulation was observed 5 days after germination, at which time no *WRKY53* or *WRKY70* mRNA was detectable. The presence of low levels of SA resulted in a significant reduction of root length in *atr8* seedlings, whereas *wrky53* and *wrky70* mutants exhibited the opposite phenotype. Taken together, *AtR8* lncRNA participates in Pathogenesis-Related Proteins 1 (PR-1)-independent defense and root elongation, which are related to the SA response. The mutual regulation of *AtR8* lncRNA and *WRKY53/WRKY70* is mediated by Nonexpressor of Pathogenesis-Related Gene 1 (NPR1).

## 1. Introduction

Non-coding RNA (ncRNA) generally refers to transcripts that do not encode proteins, and such RNAs are widely encountered in organisms. The large-scale systematic annotation and functional characterization studies of genes available in the ENCODE and FANTOM databases report that at least 80% of mammalian genomic DNA is actively transcribed into huge numbers of ncRNAs [1,2]. For plants, a genome-wide search for ncRNAs has been previously performed in a variety of plants such as *Arabidopsis thaliana*, *Medicago truncatula*, and so on [3]. These can be classified into two groups according to length: small ncRNAs (sncRNAs) and long ncRNAs (lncRNAs). SncRNAs are <200 nt in length and thus encompass micro RNAs (miRNA; approximately 22 nt), small interfering RNAs (siRNA; 20–25 nt) and related ncRNAs (piRNA, natsiRNA, and ta-siRNA; 21–30 nt), 5S rRNA (approximately 120 nt), and tRNAs (75–95 nt). Small nuclear RNAs (snRNA) and small nucleolar RNAs (snoRNA) are also classified as sncRNAs.

By contrast, lncRNAs are ≥200 nt in length. Plant lncRNAs play roles in responses to biotic and abiotic stress [4,5,6]. For example, 125 lncRNAs involved in powdery mildew infection and high-temperature stress have been identified in wheat during responses to biotic stress [7]. After exposure of *Arabidopsis thaliana* to drought, cold, high salt, or abscisic acid, the expression of 1832 lncRNAs was up-regulated markedly when compared with the control group [8]. More recently, genome-wide studies showed that lncRNAs respond to heat and salt stress in Chinese cabbage *(Brassica rapa)* and poplar (*Populus trichocarpa*), respectively [9,10].

Several lncRNAs have been identified and functionally characterized in plants [11,12,13,14,15]. The plant lncRNAs identified to date are mainly transcribed by RNA polymerase II (Pol II). In addition, a small number of plant lncRNAs have been identified that are transcribed by other RNA polymerases. These include *7-2/MRP* RNA, *U3* snoRNA, and SINE RNAs, which are transcribed by RNA polymerase III (Pol III), and scaffold lncRNAs for Argonaute binding to chromatin, which are transcribed by RNA polymerase V (Pol V) in Arabidopsis [16].

*AtR8* lncRNA in Arabidopsis was identified by in silico prediction and subsequently validated as an approximately 260 nt transcript by in vitro transcription assays involving Pol III activity in tobacco nuclear extracts [17]. *AtR8* transcription is responsive to hypoxic stress and SA treatment, and the *AtR8* lncRNA is expressed in relative abundance in the root tips of seedlings and cytosol of Arabidopsis MM2d cultured cells. *AtR8* lncRNA is conserved in six additional taxa of Brassicaceae, and the secondary structure of these RNAs is also conserved across the six taxa [17]. The closest homolog of *AtR8* lncRNA (*BoNR8* lncRNA) found in cabbage (*Brassica oleracea*) affected seed germination and growth [18]. Unlike pre-miRNAs or *ENOD40* transcripts [19], *AtR8* lncRNA was not processed into a smaller fragment, and a short open reading frame was not found in the transcribed region.

Plants are sessile organisms and are thus constantly exposed to a wide range of environmental abiotic and biotic stressors. Therefore, plants have evolved several active defense/response mechanisms to withstand adverse environmental conditions. Salicylic acid (SA) is a phytohormone and an endogenous signaling molecule that affects physiological processes such as development, photosynthesis, transpiration, ion uptake/transport, and response to stress. SA also induces specific morphological changes in the structures of leaves, roots [20], and chloroplasts. Significantly, SA signaling in plants plays a major role in mediating defense responses against microbial pathogens and herbivores [21].

Flowering plants contain many regulators in the SA signaling cascade and, among these, WRKY transcription factors (TFs) play roles of pivotal importance. The regulatory pathways of WRKY TFs are involved in the response to biotic and abiotic stresses [22,23]. WRKYs represent one of the largest families of TFs in plants. For example, Arabidopsis contains 74 *WRKY* genes, which have been divided into three groups (I–III) based on the number of WRKY domains and the structure of the zinc-finger motifs in their encoded proteins. Group I members contain two WRKY domains and a C_2_H_2_-type zinc-finger motif, group II members contain one WRKY domain and a C_2_H_2_-type zinc-finger motif, and group III members contain one WRKY domain and a C_2_HC-type zinc-finger motif [24]. AtWRKY54 and AtWRKY70, which belong to group II, have been implicated in the response to biotic stress and the orchestration of SA and jasmonic acid (JA) signaling pathways during plant defense [25,26,27]. AtWRKY22 responds to both defense and hypoxia [28]. AtWRKY54 and AtWRKY70 act cooperatively as negative regulators of leaf senescence, which is a prolonged type of programmed cell death [29,30]. AtWRKY53 is also a key player in age-induced leaf senescence [31,32,33,34]. Finally, AtWRKY46, together with AtWRKY54 and AtWRKY70, coordinates basal resistance against the plant pathogen *Pseudomonas syringae* (*P. syringae*) [35]. *WRKY* and *PR* genes, which encode Pathogenesis-Related Proteins such as PR-1, contain a conserved ‘W-box’ as a *cis* element and are regulated by Nonexpressor of Pathogenesis-Related Gene 1 (NPR1), which is a master regulator of SA-dependent plant immunity [26,36].

In this study, a microarray-based transcriptome analysis of an *AtR8* lncRNA-defective mutant (*atr8*) under hypoxic stress revealed a tight interaction between *AtR8* lncRNA and defense functions, which were induced by low-level SA. To further understand the potential defensive role of *AtR8* lncRNA, relationships between *AtR8* and *WRKY53*/*WRKY70* in the presence of SA or upon infection with *P. syringae* were also examined. Futhermore, *AtR8* lncRNA was also involved in root elongation under low SA concentrations (20 µM). Therefore, this study proposes the importance of *AtR8* lncRNA in the post-germination development of early seedlings, which is likely to relate to defense functions in the early developmental stage of plants.

## 2. Results

### 2.1. Correlation between Transcription of AtR8 lncRNA and WRKY TF Genes in Arabidopsis Seedlings under Hypoxic Stress

*AtR8* lncRNA responds to hypoxic stress and SA treatment [17]. To further understand its function, a microarray-based differential expression analysis was performed between *AtR8* lncRNA-gene defective null mutant (*atr8*; obtained from the Salk library) and wild-type (Wt) Arabidopsis seedlings. Ten-day-old Wt and *atr8* seedlings were subjected to hypoxic conditions for 6 h and then allowed to recover for 22 h under normal conditions, as previously described [17]. Total RNA was extracted from treated *atr8* and Wt seedlings and subjected to transcriptomic analysis using the Affymetrix GeneChip platform.

In total, 434 genes were differentially expressed (≥ 2-fold change; *p* ≤ 0.05) between Wt and *atr8*. Of these, 145 genes were down-regulated and 289 were up-regulated in *atr8* compared with Wt (Appendix A). Differentially expressed genes were annotated based on gene ontologies (Appendix A) and subjected to hypergeometric distribution tests for both up- and down-regulated genes. The top 15 down-regulated categories are shown in Figure 1A, and a complete list is provided in Appendix A. Genes involved in cellular biosynthetic processes such as photosynthesis, light response, and carbohydrate synthesis were down-regulated. The Arabidopsis Information Resources (TAIR) Gene Ontology (GO) analysis [37] of down-regulated genes annotated 11 gene products as metabolic process, 10 as nuclear, 13 as plastidic, and five as mitochondrial components (Appendix A). By contrast, a large proportion of up-regulated genes was found to be related to the stress response and defense (Figure 1A, lower panel). TAIR GO annotation analysis revealed that among the up-regulated genes, 26 were categorized as stress responsive and 20 were responsive to abiotic or biotic stimuli, while 14 up-regulated genes were associated with plasma membranes and five were associated with cell wall components. The first obstacle encountered by a pathogen is the plant cell wall; thus, membrane trafficking is key to establishing a rapid defense response [38,39]. Ten up-regulated genes encoded kinases, and eight encoded signal transduction components that may participate in stress responses and defense. In total, GO annotations associated at least 122 of the 434 differentially expressed genes with one or more kinds of stress or defense-related function (Appendix A). Of these 122 genes, 104 were up-regulated in *atr8* compared with Wt. Thus, a large proportion (36%) of the 289 genes up-regulated in *atr8* were associated with stress- or defense-related functions (Appendix A).

Included in the 289 genes that were up-regulated in the *atr8* mutant line were 12 WRKY-class TF-encoding genes (Table 1). To date, 74 *WRKY* genes have been identified in Arabidopsis, indicating that one in six *WRKY* genes were affected by the *atr8* mutation. WRKY TFs are involved in biotic and abiotic stress responses as well as senescence, seed dormancy, seed germination, several developmental processes, and submergence responses [22]. The up-regulation of *WRKY53*, *WRKY46*, *WRKY54*, and *WRKY70* transcription in *atr8* was validated by reverse transcription quantitative PCR (RT-qPCR), and the results were consistent with those obtained from the microarray analysis (Figure 1B). From these results, it can be inferred that hypoxia induces *AtR8* lncRNA participation in the plant defense system, including WRKY cascade. In addition, RNA gel blot analysis using Wt, *atr8*, *wrky53*, and *wrky70* mutants growing in non-stressed condition showed that *AtR8* lncRNA was not detectable in *atr8* plants and increasing in *wrky53* plants (Figure 1C). Two relatively well-studied *WRKY* genes, *WRKY53* and *WRKY70*, exhibited approximately 8- and 4-fold up-regulation, respectively, and they were thus selected for detailed functional characterization to further understand the role of *AtR8* in the stress response. 

### 2.2. Impact of SA Exposure in atr8 and wrky Mutants on Root Elongation

Previously, we reported that *AtR8* lncRNA accumulated in the primary root apices and that its accumulation was affected by treatment with SA [17]. When Arabidopsis seedlings were treated for 18 h with 0.5 mM SA and then subjected to recovery treatment (no SA, 48 h), *AtR8* lncRNA accumulation decreased to almost half of the original control levels [17]. The effect of exogenous SA application on vegetative growth is not universal, being dependent on plant species and SA concentration [21]. Although the effect of exogenous SA application on vegetative growth is dependent on plant species and SA concentration [21], we performed comparisons of the root lengths of Wt, *atr8*, *wrky53*, and *wrky70* seedlings in the presence of different concentrations because it provides a simple and effective method to assess the effect of SA. Clear differential inhibitory effects on root elongation were observed only in a narrow window of low SA concentrations (20 µM; Figure 2, Appendix A). The inclusion of 20 µM SA (SA^plus^) resulted in diminished root elongation in Wt and further reduction of root length in *atr8* seedlings. However, in SA-free media (SA^minus^), the root lengths of *atr8*, *wrky53*, and *wrky70* were similar to those of Wt. The differential effect between Wt and *atr8* was not observed at higher SA levels (25 µM or 10 mM SA) (Appendix A). By contrast, the *wrky53* and *wrky70* mutant lines exhibited longer roots than those of Wt in SA^plus^ (Figure 2). The results showed that there was a certain correlation between *AtR8* lncRNA and WRKY53 and WRKY70 under the condition of low concentration of SA, and this correlation indicated that *AtR8* lncRNA might have an antagonistic relationship with these two WRKY proteins in an SA-dependent inhibition of primary root elongation.

With respect to root elongation, the primary roots of the *shr* mutant are especially short [40]. Arabidopsis SHR and its related SCARECROW (SCR) gene, a key regulator of radial patterning and stem cell niche specification in roots [41], have been well characterized. The mutation of *SHR* and *SCR* genes causes a disorganization of the quiescent center in root apical meristems (RAM) and loss of the maintenance activity of peripheral stem cells, leading to a short-root phenotype [42,43,44]. RT-qPCR analysis indicated that *SHR* mRNA accumulation was up-regulated in both Wt and *atr8* under SA^plus^ conditions (Figure 3), and prominent accumulation was observed in the *atr8* mutant 5 days after germination (DAG).

### 2.3. Long-Term SA Treatment and Gene Responses

To study the relationship between *AtR8* lncRNA and the SA response in detail, the accumulation of *AtR8*, *WRKY53*, *WRKY70*, defense-related *NPR1*, and *PR-1* transcripts was assayed in Wt, *atr8*, *wrky53*, and *wrky70* backgrounds by RT-qPCR. Upon continuous SA^plus^ treatment for 24 days after seeding, accumulation of the two *WRKYs*, *NPR1*, and *PR-1* was induced in Wt, whereas *AtR8* was repressed (Figure 4B). When *atr8* mutants were grown under SA^plus^ conditions, *WRKY53*, *WRKY70*, and *NPR1* transcripts accumulated to higher levels than in Wt, leading to further induction of the *NPR1*-dependent *PR-1* gene (Figure 4C). In the *wrky53* mutant, *AtR8* lncRNA levels were slightly induced under SA^plus^ conditions (Figure 4D), whereas the *WRKY70*, *NPR1*, and *PR-1* mRNA levels were similar to those in Wt, regardless of the SA conditions. The *wrky70* mutants showed not statistically significant but faint induction of *AtR8* lncRNA under continuous SA^plus/minus^ conditions. However, the SA-mediated induction in *WRKY53* mRNA accumulation was lost in the *wrky70* mutants (Figure 4E). This suggests that SA influences *WRKY53* transcript levels via *WRKY70*. Furthermore, mutations in *WRKY53* or *WRKY70* led to an induction of *AtR8* lncRNA accumulation, but it had no apparent effect on the transcript levels of *NPR1* or *PR-1*. These results suggest that *PR-1* accumulation was independent of *AtR8* accumulation, which means that the *AtR8* lncRNA might participate in an unknown PR protein-independent defense mechanism.

### 2.4. Effects of Short-Term SA Treatment on Gene Responses 

Time-course experiments were conducted to precisely determine the short-term induction effects of SA on the accumulation of individual genes. Two-week-old Wt, *atr8*, *wrky53*, and *wrky70* seedlings were dipped once into a 20 µM SA solution and incubated on fresh ½ Murashige and Skoog (MS) agar plates with 20 µM SA for 0–24 h. RT-qPCR analysis showed that *WRKY53* and *WRKY70* mRNA levels in Wt increased rapidly in response to SA (after 1 h) and then decreased gradually (Figure 5F). In the *atr8* mutant, the *WRKY70* mRNA accumulation pattern was similar to that of Wt, whereas the increase in *WRKY53* mRNA levels was less pronounced in *atr8* than in Wt (Figure 5F,G). In the *wrky53* mutant, *AtR8* lncRNA levels increased gradually to more than 2-fold after 6 h in response to SA, and then declined to approximately 60% of their peak levels after 24 h. The accumulation of *WRKY70* mRNA was induced markedly 1 h after SA treatment (Figure 5F,H). In the *wrky70* mutant, *AtR8* lncRNA was induced immediately after the SA treatment, whereas the pattern of *WRKY53* mRNA accumulation was similar to that in the Wt (Figure 5F,I). These results indicate that WRKY53 or WRKY70 act as repressors in the SA-mediated regulation of *AtR8* in the short-term presence of SA, and the responses of *WRKY53* and *WRKY70* genes under low-level SA are relatively rapid (start within one hour) and further induced in the *atr8* mutant.

### 2.5. Rapid Induction of WRKY and AtR8 Gene Expression by Pseudomonas syringae

To understand the functional relevance of the interactions between *AtR8* lncRNA and the two WRKY proteins as well as their subsequent involvement in the defense response, the responses of Wt and *atr8*, *wrky53*, and *wrky70* mutants upon infection with the plant pathogen *Pseudomonas syringae* were analyzed. Two-week-old seedlings were infected with *P. syringae* and transferred to fresh SA-free ½ MS agar plates. The *P. syringae* strain used in this study did not cause a hypersensitive response (HR) in Arabidopsis. Total RNA was extracted at 0, 0.5, 1, 6, 24, and 48 h after infection and subjected to RT-qPCR. Wt and *wrky70* plants did not show any significant change in *AtR8* transcript accumulation upon infection (Figure 6B,E as well as Appendix A). However, in the *wrky53* plants, *AtR8* lncRNA levels increased nearly 3-fold at 6 h after infection, followed by gradual decrement (Figure 6D and Appendix A). In all plants, *WRKY53* and *WRKY70* transcription was induced rapidly 30 min after infection, followed by rapid decline (Figure 6 and Appendix A). This was particularly apparent in *wrky70* plants, where *WRKY53* transcript levels increased approximately 5-fold upon infection (Figure 6E and Appendix A). Therefore, WRKY53 might play a potentially pivotal role in mediating plant defense processes during the early stages of pathogen infection and act to repress *AtR8* accumulation. Overall, the *WRKY53* and *WRKY70* responses to infection were fast and pronounced, whereas *AtR8* responses (both increase and decrease) were delayed. These results indicate that the *AtR8* lncRNA surely acts on plant defense cooperating with the WRKYs-PR-1 pathway.

### 2.6. Temporal Accumulation of RNAs after Germination 

Our recent study showed that a homolog of *AtR8* lncRNA in cabbage (*BoNR8* lncRNA) affects seed germination and growth, and that its accumulation peaks 24–48 h after germination [18]. To determine whether the accumulation of *AtR8* lncRNA was temporally regulated in Arabidopsis, we performed a post-germination time-course analysis. Wt, *atr8*, and *wrky70* seeds were germinated on SA-free gellan gum gel media, and total RNAs were extracted 5, 10, and 15 DAG and subjected to RT-qPCR. As shown in Figure 7, *AtR8* lncRNA levels peaked 5 DAG and then declined gradually toward 15 DAG in both Wt and *wrky70* plants. However, the rate of decrease was lower in the *wrky70* line than in Wt. Conversely, *WRKY53* and *WRKY70* transcripts were almost undetectable 5 DAG in all plants. However, their levels increased gradually, which was concomitant with the decline of *AtR8* lncRNA levels. Moreover, increases in the levels of *WRKY53* and *WRKY70* mRNAs were higher in the *atr8* mutant than in Wt and, in this case, an inverse correlation between *AtR8* and *WRKY53*/*WRKY70* was observed. These results suggest that the *AtR8* lncRNA functions at the early developmental stage after germination, before the onset of PR protein syntheses in germinating seeds and tiny seedlings.

### 2.7. Regulation of NPR1-Mediated Genes

Since the *AtR8* and *WRKY53*/*WRKY70* genes are inversely and SA-dependently regulated, we further examined the contribution of NPR1 (a key regulator of SA-mediated immunity). For this purpose, we conducted RT-qPCR on an Arabidopsis hypomutant of *NPR1* (*npr1*) to compare *AtR8*, *WRKY53*, and *WRKY70* RNA accumulation at an early developmental stage (5 DAG; cf. Figure 7) in Wt, *atr8*, and *npr1* plants in the presence and absence of SA (Figure 8). In Wt plants, the expression of *WRKY53*, *WRKY70*, *NPR1*, and *PR-1* mRNA was induced by SA, whereas that of *AtR8* was suppressed (Figure 8B). In *atr8* plants, *WRKY53* and *WRKY70* expression was induced by SA to a greater extent than in the Wt, leading to the up-regulation of *PR-1* (Figure 8C). Although *npr1* plants were heterozygotes, almost no *NPR1* mRNA accumulation was observed; however, the levels of *AtR8* lncRNA were slightly higher in these plants than in the Wt, and the inhibitory effect of SA on *AtR8* lncRNA accumulation was lost. In *npr1* plants, *PR-1* mRNA expression was not increased by SA treatment (Figure 8C). These results suggest that the NPR1 might be a repressor of the *AtR8* gene as well as an activator of *WRKY53* and *WRKY70* genes.

## 3. Discussion

### 3.1. Involvement of AtR8 lncRNA in Plant Defense

*AtR8* lncRNA was previously shown to be abundantly accumulated in roots, with expression induced by hypoxic stress but suppressed by exogenous treatment with 0.5 mM SA [17]. Thus, hypoxia and defense are likely correlated with the activities of the WRKY network, and these factors interact with *AtR8* lncRNA and SA in the root (also see Appendix A).

Although lncRNA involvement with plant defense responses has not been widely reported [8,14,45], ELF18-INDUCED LONG-NONCODING RNA (ELENA1) lncRNA in Arabidopsis was proposed to be pathogen responsive. Its accumulation increased 22-fold after treatment with prokaryotic elongation factor thermo unstable (EF-Tu) [46], which acts as a pathogen-associated molecular pattern, although this EF-Tu induction has not been confirmed in our study. Here, we provide substantial evidence for the existence of another plant defense-related lncRNA from Arabidopsis, *AtR8*. The first piece of evidence is provided by the microarray-based transcriptome analysis of the *atr8* mutant under hypoxic/submergence stress, in which 36% of up-regulated genes (104 of 289 genes) were associated with defense responses. *WRKY* genes were strongly represented in the up-regulated genes and, as reported previously, more than 70% of the *WRKY* gene family members are responsive to defense and SA [47].

In a previous study, we explored the potential response of *AtR8* lncRNA under stress conditions; the largest changes in *AtR8* lncRNA levels were detected after waterlogging treatment [17]. Therefore, we performed a microarray-based differential accumulation analysis of Wt and *art8* seedlings subjected to hypoxic submergence conditions, as described previously [17]. WRKYs are recognized for their roles in responding to stress and defense, but some WRKYs are also thought to contribute to the submergence response. For example, the Arabidopsis *AtWRKY22* gene is involved in the regulatory networks of submergence signaling [28], while rice OsWRKY62 plays a positive role in the regulation of hypoxia-responsive genes [48]. Some aspects of the role of rice WRKYs in aerenchyma development under submergence stress were described previously [23], and Arabidopsis *AtWRKY22* and rice *OsWRKY62* were shown to respond to defense as well as hypoxia. 

In this study, a significant association was demonstrated between accumulation of the *AtR8* lncRNA and two *WRKY* genes (*WRKY70* and *WRKY53*) in Arabidopsis. WRKY70 plays a key role in integrating signals from the SA and JA pathways, with *WRKY70* accumulation activated by SA and repressed by JA [27]. WRKY53 is also SA-responsive and participates in plant development and defense [34,35]. Both WRKYs were also shown to act as regulators of leaf senescence [29,30,33]. 

The conserved WRKY domain binds preferentially to a conserved core element known as the ‘W-box’, which exhibits a common (T)TGAC(C/T) motif that has been observed in the promoters of multiple stress-inducible genes, including *Pathogenesis-Related* (*PR*) and *WRKY* genes [22,47,49,50]. Accordingly, *WRKY* genes can coordinate highly complex gene cascades during the defense response. In this study, we observed an inverse correlation between the expression patterns of *AtR8* lncRNA and *WRKY53/WRKY70* mRNA in response to SA, *P. syringae* infection, and early seedling development (Figure 2 and , Figure 4, Figure 5, Figure 6 and Figure 7), suggesting that *AtR8* lncRNA might mutually regulate WRKY functions. Previously, several attempts were made to generate double mutants of *atr8 wrky53* and *atr8 wrky70*, but the resultant heterozygotes were less likely to survive in soil, and homozygotes of these mutants were not obtained. These results suggest that such double mutants might be highly sensitive to infections and suffer from increased lethality. Importantly, *AtR8* lncRNA accumulation was found to be independent of *PR-1,* because *PR-1* mRNA levels were not changed regardless of *AtR8* lncRNA levels in *wrky53* and *wrky70* (Figure 4D,E), and *AtR8* lncRNA might therefore contribute to a *PR-1-*independent defense mechanism. This is critically different from ELENA1 lncRNA, as ELENA1 positively regulates *PR-1* gene accumulation in cooperation with the MED19a mediator protein [46]. Moreover, although binding proteins have not yet been identified, it is likely that *AtR8* lncRNA forms ribonucleoproteins. Therefore, the next step will be to analyze complexes that include *AtR8* lncRNA. 

### 3.2. Delayed Root Elongation in atr8 Strongly Associates with SA

Usually, mutant phenotypic characterization provides sufficient information to determine the functions of an uncharacterized gene. However, unlike protein coding genes, many ncRNA mutants do not display any clear phenotype. Previously, we reported that high levels of *AtR8* lncRNA accumulated in primary and lateral root apices in Wt, but that no phenotype was found in the *atr8* mutant [17]. However, careful examination of *atr8* during the course of this study revealed delayed primary root elongation in the presence of low levels of SA (20 µM; Figure 2 and Appendix A). SA plays diverse roles during plant development and in response to stress [21], and its effect can be estimated by observing the suppression of root elongation [20]. In *atr8*, root elongation was more severely affected upon the application of low levels of SA. SHORT-ROOT (SHR) protein is a DNA-binding transcription factor that promotes root elongation during the maintenance of stem cells in RAM. Microarray analysis showed that *SHR* accumulation was down-regulated in *atr8* (Appendix A), and RT-qPCR results indicated that *SHR* accumulation was induced by SA in both Wt and *atr8* plants (Figure 3). Paradoxically, SA inhibited root elongation regardless of *SHR* up-regulation. Furthermore, *SHR* induction was more pronounced in *atr8*. The reasons for this are not readily apparent; however, the maintenance of meristems is both complex and critical and may involve hitherto unidentified regulatory cascades. It is possible that *SHR* gene up-regulation at least may act to counterbalance SA-induced root shortening. However, further research is needed to fully understand this relationship. Conversely, the SA-dependent inhibitory effect on root elongation was suppressed in both the *wrky53* and *wrky70* mutants. Therefore, we assume that *AtR8* lncRNA negatively regulates the sensitivity of SA (20 µM) on root elongation; therefore, *atr8* shows a shorter root phenotype comparing with Wt. On the other hand, WRKY53 and WRKY70 positively regulate the sensitivity of SA (20 µM), and longer root phenotypes appear in those mutants. 

### 3.3. Concentration-Dependent SA Effects

The most intensively studied role of SA is in the plant immune response against biotrophic or hemibiotrophic pathogens. Generally, land plants have two layers of SA-induced active defense responses that help counter challenges from pathogens: (1) the hypersensitive response (HR) and (2) systemic-acquired resistance (SAR). These responses require different levels of SA induction. HR is induced by higher SA levels and causes rapid oxidative bursts and programmed cell death at the site of infection. SAR is longer lasting, persisting for several weeks, and it is induced by lower SA levels. Although the range of basal SA levels among plant species is diverse, basal SA levels in Arabidopsis are generally in the range of 0.25–1 µg g^-1^ fresh weight (approximately 2–7 µM), and these levels increase by a factor of at least 100 at the site of infection [51,52,53]. Several reports linking the exogenous application of SA to the defense response have been published, with conflicting results. Relatively high SA concentrations were used in the majority of studies. For example, using 5 mM SA, Li et al. (2004) reported a gradual reduction of *WRKY70* accumulation during 2–24 h after exposure [27]. Hu et al. (2012) reported peak *WRKY70* accumulation 8 h after treatment with 2 mM SA [35]. However, Shim et al. (2013) observed a gradual increase in *WRKY70* accumulation for 1 h in response to treatment with 1 mM SA [54]. Furthermore, *PR-1* accumulation increased 5 h after treatment with 5 mM SA [27]. The experimental conditions in this study involved much lower SA concentrations. Root elongation in *atr8* was restricted to 20 µM, and opposing effects were observed in *wrky70* and *wrky53* mutants for the same SA concentration (Figure 2 and Appendix A). Recently, our collaborators showed that SA (15 µM) was able to induce *AtR8* lncRNA accumulation in seeds 36 h after germination, whereas a lower SA concentration (5 µM) was found to be inhibitory [55]. In their entirety, these results suggest that the accumulation of *AtR8* lncRNA responds to low SA concentrations in a very narrow concentration window (15–20 µM). These results are corroborated by the inverse correlation observed between the expression patterns of *AtR8* lncRNA and *WRKY53/WRKY70*. This also evokes the possibility of the existence of a third, as yet unknown, type of defense response involving *AtR8* lncRNA that is distinct from the HR- or SAR-type responses. The regulation of *AtR8* lncRNA accumulation in a *PR-1*-independent manner supports this hypothesis (Figure 4).

### 3.4. How are AtR8 and WRKY70 Expression Regulated by SA?

As mentioned above, the accumulation of *AtR8* and *WRKY53/WRKY70* transcripts correlated inversely, i.e., the deletion of one increased the expression of the other. Therefore, the next logical question is: How is this expression regulated? Here, we propose a plausible model in which NPR1 regulates the accumulation of *AtR8* and *WRKY53/WRKY70*.

NPR1, a key regulator of basal and SAR and SA signaling in plants, confers immunity through a transcriptional cascade (which includes transcription activators and repressors), leading to a massive induction of antimicrobial gene expression. In the *npr1* mutant, SA-induced transcriptional reprogramming, including SAR, is almost lost [26,36]. Although NPR1 cannot bind to an SA molecule directly, its paralogs (NPR3 and NPR4) are SA receptors, and they bind to SA with different affinities and regulate the ubiquitination of NPR1 [56]. In the absence of SA, oligomerized NPR1 proteins are retained in the cytoplasm, but SA-activated NPR1s are reduced together with cellular redox changes, monomerized, and translocated to the nucleus. Monomeric NPR1 does not exhibit DNA-binding activity as a TF; instead, it regulates target genes by interacting with a wide range of transcription activators, repressors, and cofactors that include members of the WRKY and bZIP transcription factor TGA families as well as NIM1-interacting proteins (NIMINs) [57,58]. NPR1 was proposed to regulate the transcription of *PR* genes together with the TGA family TF as a cofactor [59]. Numerous previous studies showed that NPR1 up-regulated *WRKY70* accumulation in an SA-dependent manner [27,60]. How is *AtR8* and *WRKY70*/*WRKY53* accumulation regulated inversely? To answer this question, we hypothesize an NPR1-mediated model. First, the NPR1/TGA complex could bind to *WRKY* genes on the W-box and to the *AtR8* gene; this is because W-box-like sequences reside in their 5′ proximal regions. Second, we think that NPR1/TGA might regulate those genes inversely. To evaluate this hypothesis, we generated *NPR1* mutants (*npr1*) (Figure 8). At an early growing stage (5 DAG; cf. Figure 7), a defect in the *NPR1* gene (*npr1*) contributed to the accumulation of *AtR8* lncRNA, regardless of the SA conditions, thereby confirming this theory. By contrast, deregulation of NPR1 in the *npr1* mutant plant does not require the *de novo* production of accessory proteins and can thus proceed. Even though the proposed mechanism seems feasible, further investigations into the mechanism underlying the NPR1-mediated regulation of *AtR8* lncRNA are required for confirmation.

### 3.5. Biological Implications of the Involvement of AtR8 lncRNA in the Defense Response

Although *AtR8* lncRNA accumulates predominantly in the roots, it was recently shown to also accumulate in germinating seeds [55]. One of the common keywords that links roots and seeds is ‘soil’. As many plant diseases originate in the soil, plants are constantly required to select alternative strategies based on a ‘growth–defense trade-off’ [61,62]. For germinating seeds, growth should be prioritized over other physiological processes, but investments in defense are also essential to neutralize the challenges posed by surrounding pathogens. However, because resources are limited within the relatively small Arabidopsis seed, an optimal defense response that is frugal in energy consumption would be preferred at this stage. HR and SAR are highly evolved and finely tuned systems, but their immense metabolic cost would undoubtedly be disadvantageous for fragile seedlings in their early stages of development. Our data suggest that an additional primitive, energy-efficient, defense mechanism involving *Atr8* lncRNA may be present.

If this kind of defense mechanism exists, the participation of reactive oxygen species (ROS) might not be negligible, because several pieces of collateral evidence link *AtR8* lncRNA to ROS production. *AtR8* lncRNA accumulation in Wt seeds was enhanced upon the application of SA [55]. In general, SA induces the production of ROS during seed germination, and ROS play a key role during seed germination and protection against pathogens [63,64]. As ROS can be generated in mitochondria, chloroplasts, or peroxisomes in the cytoplasm, the localization of *AtR8* lncRNAs in the cytoplasm correlates well with the site of ROS production. Interestingly, the accumulation of many mitochondria- and chloroplast-related genes was also significantly down-regulated in *atr8* mutants (Appendix A). As roots are the first point of contact with the microbial community in the soil, defense-related proteins are constitutively expressed in roots [65,66]. However, an *AtR8*-based defense mechanism could be operative during the early stages of root development to tilt the balance in favor of an energy trade-off. Our data clearly demonstrate that *atr8* mutation causes suppressed root elongation, while *wrky70* and *wrky53* mutation resulted in enhanced root elongation. In parallel, the proposed PR-1-independent defense mechanism might be suppressed in the *atr8* mutant but enhanced in the *wrky70* and *wrky53* mutants. 

Under low SA concentrations, deletion of the *WRKY53* gene did not affect *WRKY70* accumulation; however, the loss of *WRKY70* did affect *WRKY53* gene accumulation (Figure 4 and Figure 5), suggesting that *WRKY70* may act upstream of *WRKY53* in a WRKY cascade (Figure 9). 

In conclusion, the data presented in this paper strongly suggest the involvement of *AtR8* lncRNA in a defense response that also involves WRKY70/WRKY53 (Figure 9). At a minimum, the defense mechanism operates in the roots and at the early stages of seed germination. Further studies are required to elucidate the mechanistic details underlying this response.

## 4. Materials and Methods

### 4.1. Plant Materials and Growth Conditions

*Arabidopsis thaliana* (L.) Heynh. accession Wassilevskija (WS) and *AtR8* lncRNA-defective mutant FLAG-410H04, designated *atr8* (derived from WS by T-DNA insertion), were obtained from the Versailles Arabidopsis Stock Center (the National Institute for Agricultural Research, Versailles, France, http://publiclines.versailles.inra.fr/). T-DNA insertion mutants *wrky53* (SALK_034157c), *wrky70* (SALK_025198c), and *npr1* (SALK_204100c) (derived from Arabidopsis accession Columbia) were obtained from The Arabidopsis Biological Resource Center (ABRC, the Ohio State University, Clumbus, OH, USA, https://abrc.osu.edu/). All mutants (except for *npr1*) used in this study harbor null alleles. Mutants were characterized by PCR and RT-qPCR. Double mutants *atr8 wrky53* and *atr8 wrky70* were obtained by crossing *atr8* with *wrky53* or *wrky70*, respectively. Arabidopsis plants were maintained under a 16 h light (approximately 60 µmol m^-2s^)/8 h dark cycle at 20 °C. Seedlings were grown on soil or on half-strength (½) Murashige and Skoog (MS) medium supplemented with 2% (*w*/*v*) sucrose, 0.5 µg mL^− 1^ thiamine-HCl, 0.2 mM phosphate buffer pH 6.6, and 0.24% (*w*/*v*) or 0.4% (*w*/*v*) gellan gum.

### 4.2. Transcription Analysis of atr8 Mutant

Wild-type (Wt) and *atr8* seedlings (10 DAG) were dipped in air-purged water for 6 h (hypoxic stress treatment) and then placed in fresh ½ MS agar medium for recovery. Total RNA was extracted using Tripure Isolation Reagent (Roche Life Science, Penzberg, Germany) 22 h after the recovery treatment. Total RNA was used for microarray analysis using an Affymetrix ATH1 Arabidopsis expression array. 

Total RNA (up to 500 ng) isolated from *atr8* and Wt seedlings (10 DAG) was amplified and labeled using a 3′ IVT Express kit (Affymetrix, Santa Clara, CA, USA). The target preparation, hybridization, washing, staining, and scanning of array chips were carried out in accordance with manufacturer protocols. Affymetrix GeneChip Command Console^®^ 3.0 (AGCC) software was used to control the washing and staining of array chips in a Fluidics Station 450 (Affymetrix) as well as scanning with a Scanner 3300 (Affymetrix). Three biological replicates, processed at each stage of microarray analysis and with overall correlation coefficient values of ≥ 0.95, were used for final data analysis. Differential expression analyses for mutant lines were performed on quantile-normalized datasets with a fold change cut-off = 2 at *p* ≤ 0.05 using the CLC Main Workbench 7.8.1 (CLC Bio, Aarhus, Denmark, https://www.qiagenbioinformatics.com/) software package. Microarray data are deposited in the Gene Expression Omnibus database (NCBI, Bethesda, MD, USA).

### 4.3. Gene Expression Analysis

Total RNA was extracted from whole seedlings using Tripure Isolation Reagent. Reverse transcription quantitative PCR (RT-qPCR) was performed using a One-Step SYBR PrimeScript RT-PCR Plus Kit (Takara Bio) and the Eco Real-Time PCR System (Illumina) in 10 µL reactions containing 50 ng or 100 ng of total RNA. Reverse transcription was carried out at 42 °C for 5 min, followed by 40 cycles at 95 °C for 5 s and 60 °C for 30 s. A comparative quantitation was carried out via the 2^(−∆Ct)^ method or 2^(−∆∆Ct)^ method [67]. Each amplified fragment was validated by electrophoresis and by sequencing. The primers used for RT-PCR are listed in Appendix A.

For northern analysis, the RNA probe template, plasmid YA41-R8, was constructed using a genomic PCR fragment containing the *AtR8* lncRNA transcribed region, which was amplified with the following primers: 5′ CGG TCT AGA GGG GTG TGG GAA CCT AGG AGA 3′ (forward) and 5′ ATT GGA TCC GAG GAA ACG GTT AAC CGC AGA 3′ (reverse). Total RNA from seedlings (5 µg) was separated on a 6% (*w*/*v*) polyacrylamide gel (7 M urea, 1× Tris-Borate-EDTA (TBE) buffer) and transferred onto a nylon membrane (Hybond-N, GE Healthcare). After UV cross-linking and pre-hybridization with DIG Easy Hyb (Roche Life Science), the membrane was hybridized overnight at 45 °C with a 1 µg/mL DIG-labeled riboprobe in DIG Easy Hyb. The membrane was washed twice with 2× Saline-Sodium-Citrate (SSC) buffer/0.1% (*w*/*v*) SDS at room temperature followed by washing with 0.2× SSC/0.1% (*w*/*v*) SDS at 42 °C. After blocking with 1.5% (*w*/*v*) blocking reagent (Roche Life Science) and treatment with anti-DIG Alkaline Phosphatase (AP)-conjugated antibody (1:10,000, Roche Life Science), the blots were detected using CDP-*star* (Roche Life Science) using Hyper film ECL (GE Healthcare).

### 4.4. Salicylic Acid Treatment

Seedlings from Wt as well as *atr8*, *wrky53*, and *wrky70* mutants were germinated on a horizontal plane of ½ MS medium with 0.24% (*w*/*v*) gellan gum, with or without 20 µM SA. At 10 DAG, seedlings were transplanted onto a vertical plane of ½ MS medium with 0.4% (*w*/*v*) gellan gum, with or without 20 µm SA, and grown for 10 more days. The length of the primary root was measured using the ImageJ [68] software package.

Time-course experiments for SA treatment were performed as follows: seedlings (Wt, *atr8*, *wrky53*, and *wrky70*) were grown on ½ MS medium with 0.24% (*w*/*v*) gellan gum for 2 weeks, dipped in 20 µM SA solution (or water as a control), and returned to fresh ½ MS medium with (20 µM) or without SA (control). Total RNA was extracted 0, 0.5, 1, 6, and 24 h after SA treatment.

### 4.5. Infection with Pseudomonas Syringae 

Arabidopsis seedlings were infected with the plant pathogen *Pseudomonas syringae* pv. tagetis (*P. syringae*) as follows: *P. syringae* was streaked and cultured on King’s agar medium for 2 days at room temperature and then cultured overnight at 37 °C in King’s liquid medium. An aliquot (500 µL) of culture was centrifuged, and the bacterial pellet was suspended in 50 mL sterile water to generate a pathogen suspension. Two-week-old seedlings grown on ½ MS medium with 0.24% (*w*/*v*) gellan gum were dipped in pathogen suspension and then placed onto a fresh ½ MS plate. Seedlings were harvested 0, 0.5, 1, 6, and 24 h after infection, and total RNA was extracted from 10 seedlings at each time point.

## Figures and Tables

**Figure 1 ncrna-06-00008-f001:**
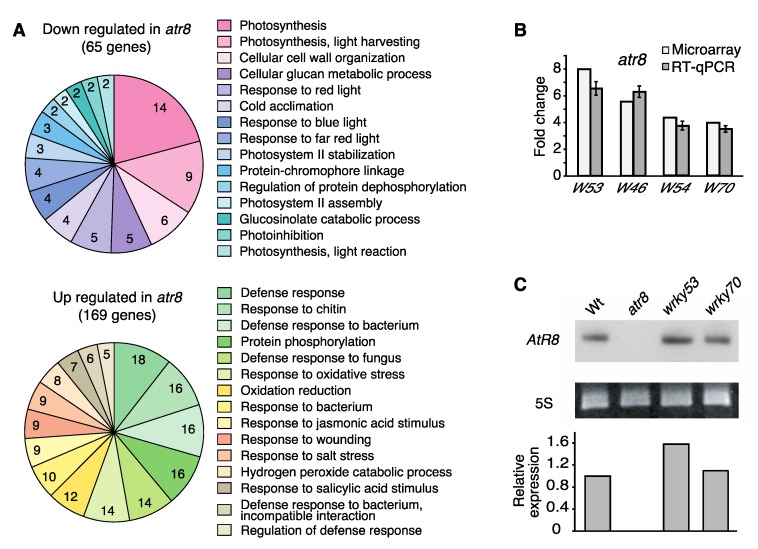
Microarray analysis and *WRKY* genes accumulation in *atr8* mutant. (**A**) Microarray analysis using the total RNAs from *atr8* and wild-type (Wt) in combination with Affimetrix GeneChip (ATH1 Arabidopsis expression array). Function categories of down- and up-regulated genes were obtained by a hypergeometric distribution test (cf. Appendix A). Each of the top 15 categories are shown on a pie chart with the number of genes classified. (**B**) Comparison of WRKY transcription factor (TF) gene accumulation in *atr8* mutant dipped in water for 6 h and then recovered for 24 h. These mRNA accumulation levels were determined by GeneChip and RT-qPCR with the same RNA samples, while *ACT2* and *ACT8* genes were used as standards for RT-qPCR. Bar graphs represent the mean values of three independent assays, and the error bars represent ±SE. *W53*: *WRKY53*, *W46*: *WRKY46*, *W54*: *WRKY54*, and *W70*: *WRKY70.* (**C**) RNA gel blot analysis showing the basal expression of *AtR8* lncRNA in Wt as well as the *atr8*, *wrky53*, and *wrky70* mutant treated with water as (**A**). The quantification of the gel blot was made using ImageJ.

**Figure 2 ncrna-06-00008-f002:**
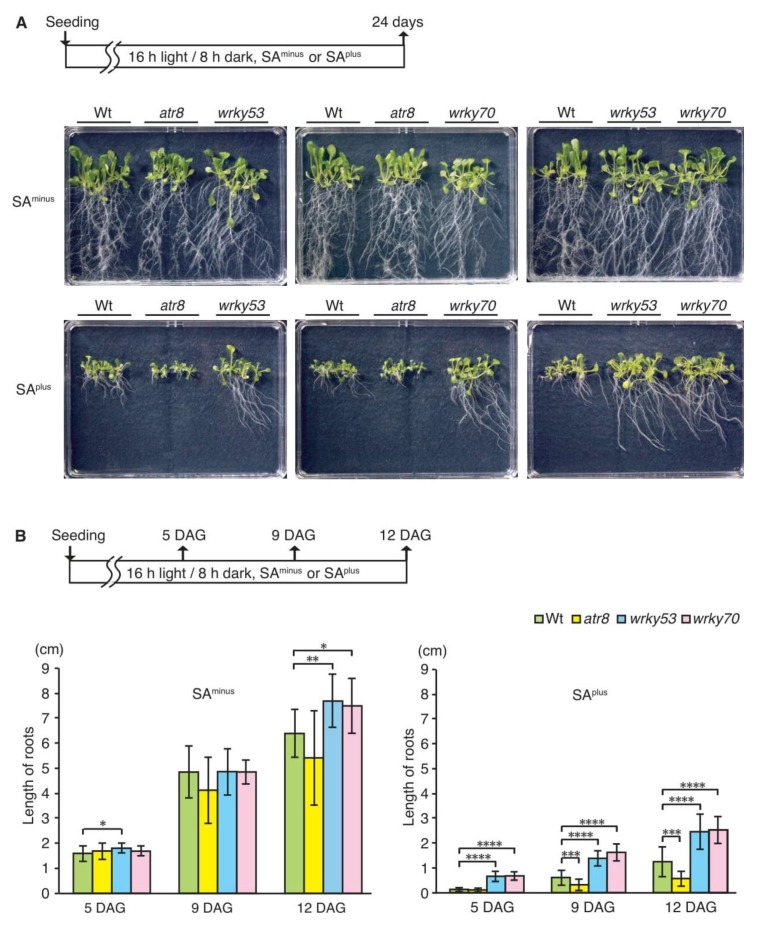
Inhibitory effects on the root elongation under lower salicylic acid (SA) conditions. (**A**) The root elongations of Arabidopsis seedlings were compared, which were grown on vertical 0.24% gellan gum medium with (SA^plus^) or without (SA^minus^) 20 µM SA for 24 days after seeding. To eliminate gellan gum plate-dependent difference, Wt and two out of the three mutant plants were combined with three individual plates. (**B**) The lengths of the primary root of (Appendix A) was measured using ImageJ. Seedlings were grown (**A**) and root lengths were compared at 5, 9, and 12 days after germination (DAG). Bar graphs represent the mean values of more than 10 plants, and error bars represent ±SD. Significant differences in the lengths of the primary root between Wt and each of the mutants are indicated as * *p* < 0.05, ** *p* < 0.01, *** *p* < 0.001, and **** *p* < 0.0001 (Welch’s *t*-test).

**Figure 3 ncrna-06-00008-f003:**
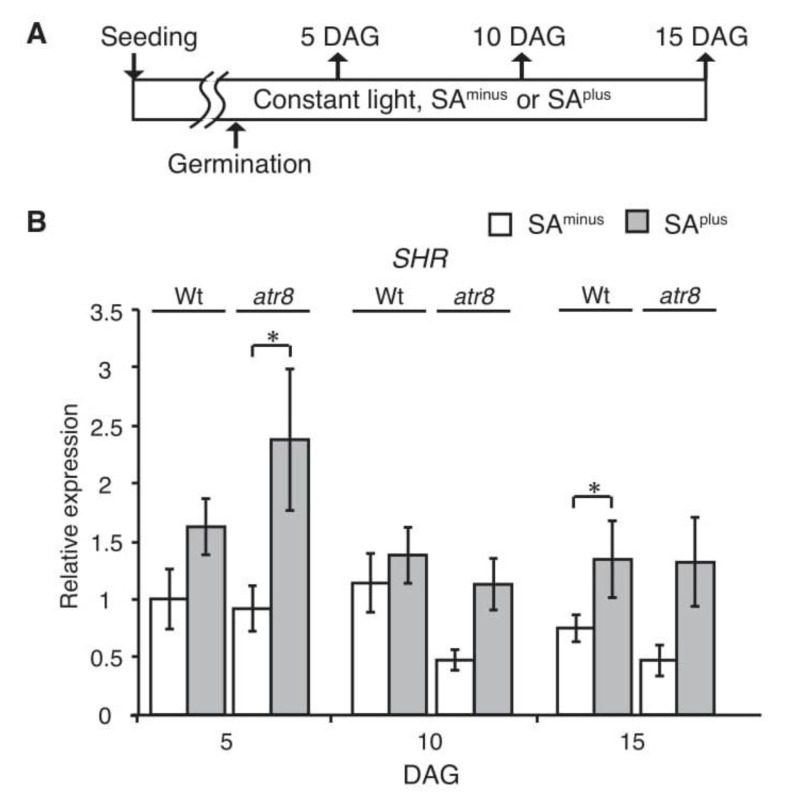
Accumulation levels of *SHR* mRNAs in Wt and *atr8* plants soon after germination. (**A**) Timeline of experimental procedure. Up arrows indicate samplings. (**B**) Time course of *SHR* mRNA accumulations. Arabidopsis seedlings that were grown on vertical 0.24% gellan gum media with (SA^plus^) or without (SA^minus^) 20 µM salicylic acid, and total RNAs were extracted at 5, 10, and 15 days after germination (DAG), and subjected to RT-qPCR with *ACT2* and *ACT8* genes as standards. Bar graphs represent the mean values of three independent assays, and the error bars represent ± SE. Significant differences in RNA accumulation levels between SA^minus^ and SA^plus^ are indicated as * *p* < 0.05 (Welch´s *t*-test).

**Figure 4 ncrna-06-00008-f004:**
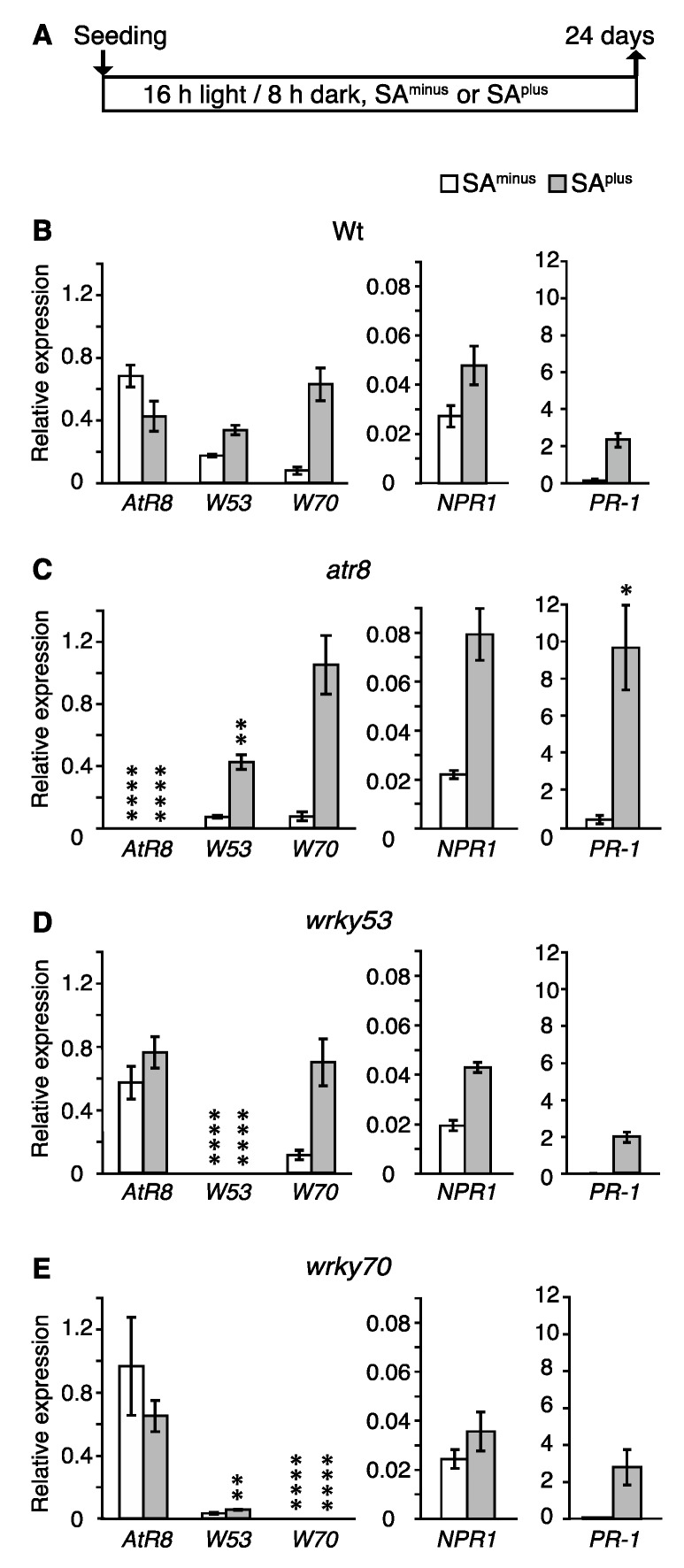
Accumulation levels of RNAs in Arabidopsis under continuous treatment with 20 µM SA. (**A**) Timeline of experimental procedure. Up arrow indicates sampling. The total RNAs were extracted from the same seedlings shown in Appendix A (24 days after seeding). (**B**–**E**) The mRNA accumulation levels SA^minus^ or SA^plus^ were compared by RT-qPCR with *ACT2* and *ACT8* genes as standards. Bar graphs represent mean values of three independent assays (≥50 plants), and error bars represent ±SE. *W53*: *WRKY53*, *W70*: *WRKY70*. Significant differences in RNA accumulation levels between Wt (**B**) and each of the mutants (**C**–**E**) are indicated as * *p* < 0.05, ** *p* < 0.01 and **** *p* < 0.0001 (Welch´s *t*-test).

**Figure 5 ncrna-06-00008-f005:**
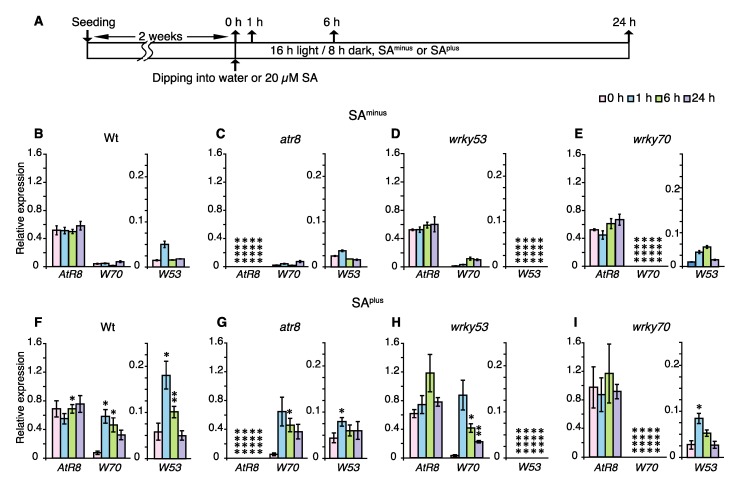
Short-term time course of RNA accumulation immediately after treatment with SA. (**A**) Timeline of experimental procedure with (SA^plus^) or without (SA^minus^) 20 µM salicylic acid. Two-week-old seedlings were dipped once into the water or 20 µM SA and incubated on fresh media (SA^minus^ or SA^plus^). Up arrows indicate samplings. Accumulations of *AtR8* lncRNA as well as *WRKY53* and *WRKY70* mRNA in Arabidopsis plants were measured by RT-qPCR with *ACT2* and *ACT8* genes as standards. (**B**–**E**) Controls that were treated with just water (SA^minus^). (**F**–**I**) Seedlings treated with 20 µM SA (SA^plus^). Bar graphs represent mean values of three independent assays (≥ 50 plants), and error bars represent ±SE. *W53*: *WRKY53*, *W70*: *WRKY70*. Significant differences in RNA accumulations levels between the same strain SA^minus^ and SA^plus^ (Wt: B and F, *atr8*: C and G, *wrky53*: D and H, and *wrky70*: E and I) are indicated as * *p* < 0.05, ** *p* < 0.01 and **** *p* < 0.0001 (Welch´s *t*-test).

**Figure 6 ncrna-06-00008-f006:**
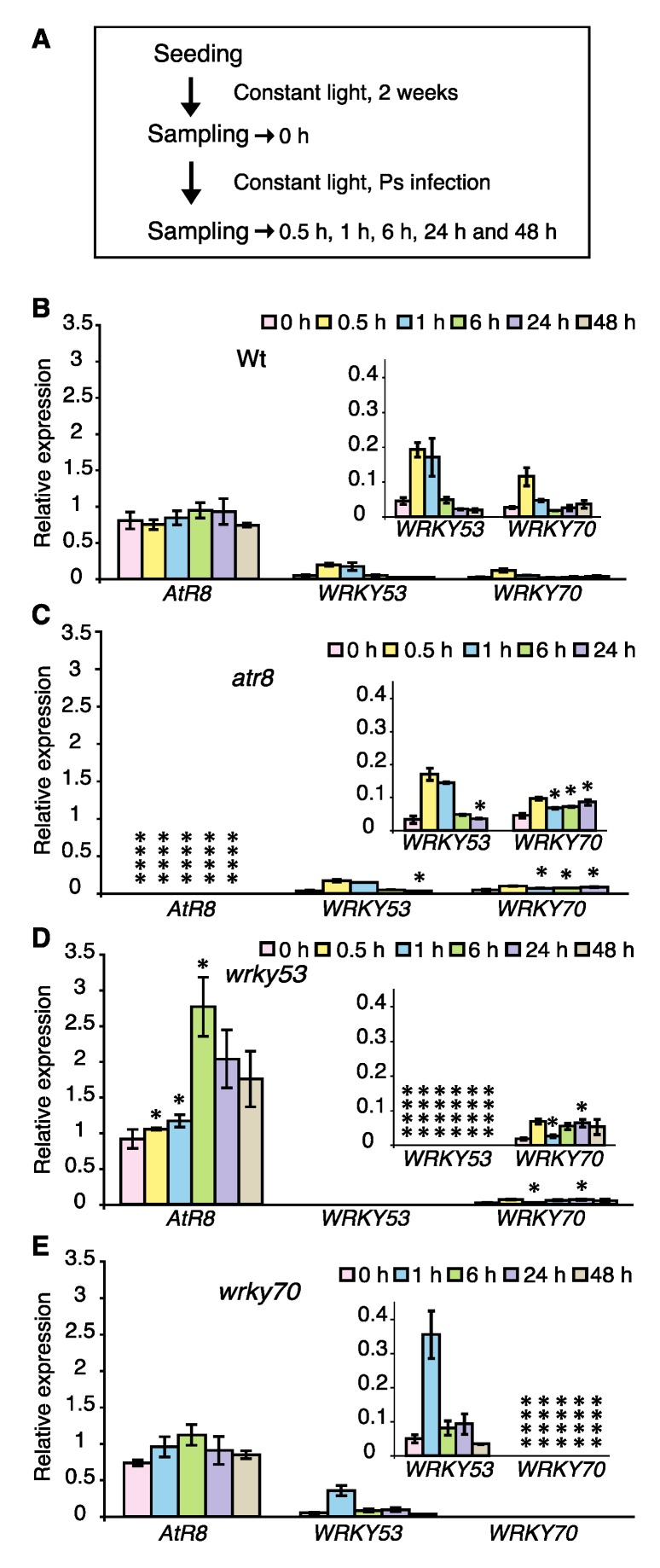
Short-term change of RNA levels by infection of plant pathogen *Pseudomonas syringae* (Ps). (**A**) Timeline of experimental procedure. *AtR8* lncRNA, *WRKY53*, and *WRKY70* in *Arabidopsis* plants were treated with Ps, which was cultured with 500 µl of King´s liquid medium and suspended in 50 mL of sterile water (100 times dilution). RNA accumulation levels of *AtR8* lncRNA, *WRKY53*, and *WRKY70* mRNAs with Ps were compared by RT-qPCR with *ACT2* and *ACT8* genes as standards. Bar graphs represent the mean values of three independent assays (more than 50 plants), and error bars represent ±SE. The X-axis magnified bar graphs are superimposed in each graph. Significant differences in RNA accumulations levels between Wt (**B**) and each of the mutants (**C**–**E**) are indicated as * *p* < 0.05 and **** *p* < 0.0001 (Welch´s *t*-test).

**Figure 7 ncrna-06-00008-f007:**
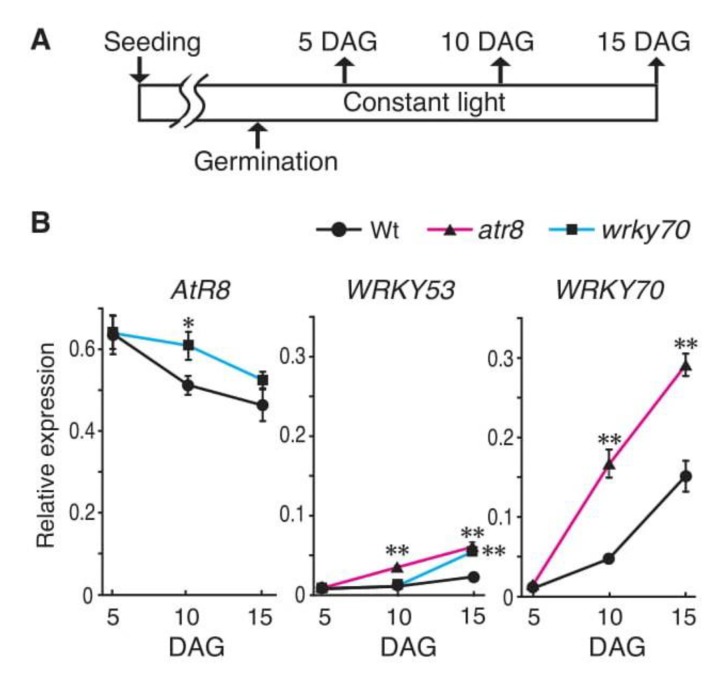
Time course of the RNA accumulations soon after germination. (**A**) Timeline of the experimental procedure. Up arrows indicate samplings. The Wt, *atr8,* and *wrky70* strains were germinated on 0.24% gellan gum media (without SA and *P. syringae)*, and the total RNAs were extracted after the indicated periods (DAG: days after germination) from each strain, and subjected to RT-qPCR with *ACT2* and *ACT8* genes as standards. (**B**) Line graphs represent the mean values of three independent assays (≥ 60 plants), and error bars represent ±SE. Significant differences in RNA accumulation levels between Wt and each of the mutants (Wt and *atr8,* Wt and *wrky70)* are indicated as * *p* < 0.05 and ** *p* < 0.01 (Welch’s *t*-test).

**Figure 8 ncrna-06-00008-f008:**
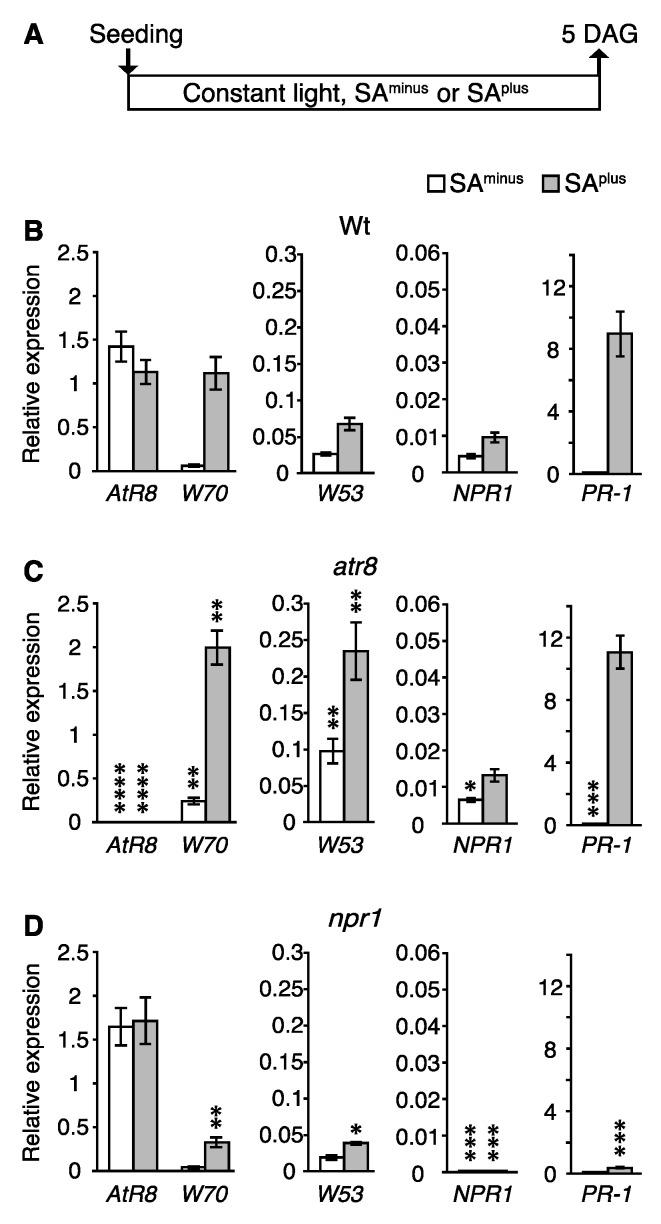
RNA accumulation levels in Arabidopsis plants including *npr1* at an early developmental stage. (**A**) Timeline of experimental procedure. The total RNAs were extracted from the seedlings of Wt, *atr8,* and *npr1*, which were grown on 0.24% gellan gum media SA^plus^ or SA^minus^ (20 µM SA), and subjected to RT-qPCR with *ACT2* and *ACT8* genes as standards. Bar graphs represent the mean values of three independent assays (≥ 100 plants), and error bars represent ±SE. *W53*: *WRKY53*, *W70*: *WRKY70*. (**B**) Wt, (**C**) *atr8* (*AtR8* null mutant), and (**D**) *npr1* (*NPR1* hypomutant). Significant differences in RNA accumulation levels between Wt (**B**) and each of the mutants (**C**–**D**) are indicated as * *p* < 0.05, ** *p* < 0.01, *** *p* < 0.001, and **** *p* < 0.0001 (Welch’s *t*-test).

**Figure 9 ncrna-06-00008-f009:**
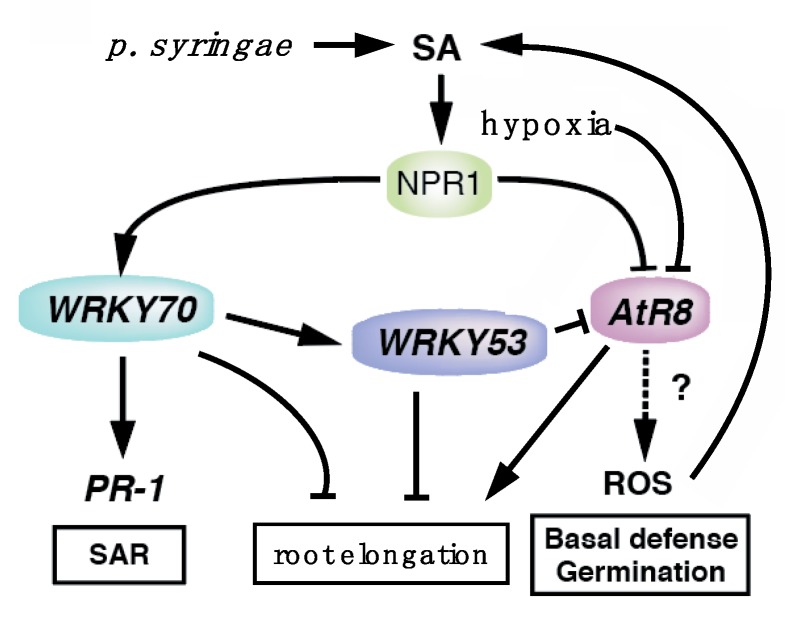
Proposal working model that illustrates the interaction among the *AtR8* long non-coding RNA (lncRNA) and WRKY network with SA-dependent defense signaling. *AtR8* lncRNA might regulate the basal defense system independent from SAR. The *AtR8* lncRNA potentially exhibits an inverse correlation to the *WRKY70* gene with SA-dependent defense signaling and root elongation. *WRKY70* is upstream in the WRKY signaling cascade, while *AtR8* lncRNA is repressed by *WRKY53*. The *NPR1* might be a key regulator for mutual regulation between *AtR8* lncRNA and *WRKY70* genes.

**Table 1 ncrna-06-00008-t001:** WRKY transcription factor genes that exhibit a more than 2-fold change in *atr8*.

TAIR ID	Fold Change	Regulation	Expression Level	Gene Title
AT1G80840	9.87	up	1137.8	WRKY40
AT4G23810	7.95	up	1595.3	WRKY53
AT2G46400	5.52	up	786.6	WRKY46
AT5G22570	5.03	up	211.4	WRKY38
AT2G40750	4.31	up	1474.1	WRKY54
AT5G13080	4.14	up	464.9	WRKY75
AT3G56400	3.94	up	3225.8	WRKY70
AT2G38470	3.39	up	1878.9	WRKY33
AT5G49520	3.08	up	475.3	WRKY48
AT2G25000	2.21	up	784.7	WRKY60
AT1G66550	2.04	up	9.4	WRKY67
AT4G31550	2.04	up	1212.9	WRKY11

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
