# Peer review of "The Arabidopsis Hypoxia Inducible AtR8 Long Non-Coding RNA also Contributes to Plant Defense and Root Elongation Coordinating with WRKY Genes under Low Levels of Salicylic Acid"

_ncrna, 2020, doi:10.3390/ncrna6010008_

Round 1
Reviewer 1 Report
This paper by Li et al. demonstrates an inverse correlation between the expression of AtR8 lncRNA and two WRKY genes (WRKY70 and WRKY53) in response to SA, P. syringae infection, and early seedling development in Arabidopsis. In general, the experiments are performed carefully, with the appropriate controls, and the data support the conclusions drawn. The manuscript is well-written and organized. The paper could be of interest to a wide audience and I would recommend publication if authors address the points described below.
- I think the authors are too conclusive in some places where the differences are not statistically significant. For example, in Figure 4E, it is stated that “the wrky70 mutants showed slight induction of AtR8 lncRNA under continuous SAplus/minus conditions”. I don’t see any significant difference compared to wild type. I suggest they could simply tone down such conclusions.
- Figure 4: When atr8 mutants were grown under SAplus conditions, WRKY53, WRKY70, and NPR1 transcripts accumulated to higher levels than in Wt, leading to further induction of the NPR1-dependent PR-1
I am failed to understand the interpretation “induction of the NPR1-dependent PR-1 gene” as there is no support from any experiment (eg. npr-1 mutant or overexpession line) to demonstrate the dependence of NPR1 for PR1 expression.
- Figure 5H: In the wrky53 mutant, AtR8 lncRNA levels increased gradually to more than 2-fold after 6 h in response to SA, and then declined gradually to ~60% of their peak levels after 24 h.
After 6 hours, only one reading (24hours) was taken, therefore I think it might not be appropriate to state “declined gradually”.
- Figure7: wrky53 mutant has consistently been used along with Wt, atr8, and wrky70 in all experiments except the time-course RNA accumulation experiment. Please provide the explanation why wrky53 mutant was not included in this experiment.
Author Response
Thank you very much for your critical review. According to your comments, the manuscript was revised as below.
Reviwer’s comment:
I think the authors are too conclusive in some places where the differences are not statistically significant. For example, in Figure 4E, it is stated that “the wrky70 mutants showed slight induction of AtR8 lncRNA under continuous SAplus/minus conditions”. I don’t see any significant difference compared to wild type. I suggest they could simply tone down such conclusions.
Answer:
As you pointed out, the difference was not statistically significant, so ‘slight’ was changed to ‘not statistically significant but faint’.
Comment:
Figure 4: When atr8 mutants were grown under SAplus conditions, WRKY53, WRKY70, and NPR1 transcripts accumulated to higher levels than in Wt, leading to further induction of the NPR1-dependent PR-1
I am failed to understand the interpretation “induction of the NPR1-dependent PR-1 gene” as there is no support from any experiment (eg. npr-1 mutant or overexpession line) to demonstrate the dependence of NPR1 for PR1 expression.
Answer:
Thank you for your comment. In general, pathogen-exposed tissues in plants result in increment of SA level and induction of the PATHOGENESIS RELATED (PR) genes. One of the key regulatory elements in SA-dependent activation of PR genes is NON-EXPRESSOR OF PR GENE 1 (NPR1). Several WRKY transcription factors are also known to play important roles, downstream of NPR1, in mediating the defense responses in plants. These explanations are found in the Introduction part (new line 136-138) as ‘ WRKY and PR genes, which encode PATHOGENESIS-RELATED PROTEINS such as PR-1, contain a conserved ‘W-box’ as a cis-element and are regulated by NONEXPRESSOR OF PATHOGENESIS-RELATED GENE 1 (NPR1), which is a master regulator of SA-dependent plant immunity [24,34]’. And as shown in Figure 8, npr-1 hypomutant experiment was already provided. In the case of npr-1, the PR-1 mRNA level was largely dropped under SA-plus condition.
Comment:
Figure 5H: In the wrky53 mutant, AtR8 lncRNA levels increased gradually to more than 2-fold after 6 h in response to SA, and then declined gradually to ~60% of their peak levels after 24 h.
After 6 hours, only one reading (24hours) was taken, therefore I think it might not be appropriate to state “declined gradually”.
Answer:
As you pointed out, after 6 hours, once tested at 24 hours, so the description was not accurate, that “declined gradually” was changed to “declined”.
Comment:
Figure7: wrky53 mutant has consistently been used along with Wt, atr8, and wrky70 in all experiments except the time-course RNA accumulation experiment. Please provide the explanation why wrky53 mutant was not included in this experiment.
Answer:
Thank you for your opinion and we have to excuse for it. It is complete our blind point why wrky53 has not been used for Figure 7 experiment. At the beginning of this experiment, the wrky53 null mutant had not been obtained, that’s why this experiment was done without the mutant. Although we cannot repeat this experiment right now within 5 days (till deadline of this revision), we still believe that the experiment in Figure 8 provides substantial evidence to prove that AtR8 lncRNA accumulates at early stage after germination and then followed by WRKY53 and WRKY70 mRNA accumulation.
Reviewer 2 Report
The paper presented for review is an attempt to determine the role of previously identified Arabidopsis thaliana AtR8 lncRNA and other elements of the plant's defense response pathways to stress (biotic and abiotic) during seedling development, root elongation and germination. I appreciate the amount of research conducted by the Authors and their efforts in the interpretation of the results. I do not know if this is just my feeling, but the results obtained are difficult to combine into a logical whole. At the beginning, microarray analyses were carried out in whole seedlings of WT and atr8 plants regarding the impact of hypoxic stress. It turned out that the different WRKY genes (in opposite manner to AtR8) may be associated with the defense response induced by hypoxic stress. The next subsection presents phenotypes of various wrky and atr8 mutants compared to WT but associated with salicylic acid (SA) dependent root elongation. Therefore, the question arises whether there is any relationship between hypoxic stress and the level of SA accumulation (regardless of the process discussed). What type of defense responses does abiotic stress cause, and is it associated (if so how) with changes in SA accumulation? Perhaps it would be worth discussing in the introduction section. Then, in various mutants, transcriptional activity of genes induced by long- or short-term SA treatment was demonstrated, but in two weeks old seedlings. Further subsections are consistent and concern the effect of P. syringae (biotic stress) on the expression of the studied genes in 14-day WT plants and mutants, as well as the temporary RNA accumulation of AtR8 and WRKY genes after germination. Nevertheless, the impact of biotic stress on the accumulation of SA should be presented. To sum up, reading the article lacks a general view on the problem, while overwhelming the huge amount of detailed data. Extensive discussion answers some of the questions, but there is no consistent combination of all results. Due to the large number of valuable results, I believe that the article is ready for publication after minor corrections and generalizations of the manuscript.
Remarks:
Due to the huge amount of results (difficult to interpret), one should write in each subsection of the “Results section” a preliminary conclusion (how the authors interpret the obtained results). The Authors wrote brief conclusions, but in my opinion they should be more complex and extended to facilitate reading of the article.
Subsection 2.1 - Based on the results presented in subsection 2.1, can we conclude that AtR8 lncRNA is a repressor of defense responses arising from hypoxia stress in A. thaliana seedlings?
Is there any explanation why hypoxic stress works in contrast to SA in the AtR8 accumulation process?
Is there somewhere in the text a discussion of Figure 1C?
L 145-148 - Please consider more than 4 genes for data validation.
Are there any other genes associated with stress/defense reaction (e.g. NPR1 or PR) among differentially expressed genes, which are taken into account in subsequent analyzes?
Subsection 2.2 - Is there any relationship to root length, SA level and plant response to biotic or abiotic stress? How can explain the inhibition of root elongation in wild-type plants after SA treatment? Additionally, based on the results of section 2.2 one could conclude that AtR8 stimulates root elongation in response to low SA levels, in contrast to the hypoxia-induced defense response?
Subsection 2.3
L 202 “suggesting that PR-1 accumulation was independent of AtR8 accumulation” - please explain because I do not understand the authors' reasoning
Subsection 2.5
In order to draw more complete conclusions, it would be worth examining the endogenous level of salicylic acid in WT plants and individual mutants, and then correlating with the transcriptional activity of the studied genes. The same applies to other analyses, e.g. studies on the impact of hypoxia stress and even those with exogenous application of SA (plants contained a certain level of endogenous SA).
Minor remarks:
Please correct the citation of papers throughout the manuscript according to the journal guidelines.
L 2-4 - In my opinion, the title can be improved because the work contains more results than the title of the work indicates.
L 16-30 - The abstract should be improved. On the one hand, it contains too detailed information (e.g. the sentence "36% of 289 up-regulated genes in atr8, including 12 SA-responsive WRKY genes, were associated with ... "), while on the other hand abbreviations not quite obvious for everyone, such as PR-1 or NPR1.
L 16 - AtR8 lncRNA was previously identified in the flowering plant Arabidopsis thaliana as..
L 21 - WRKY genes
L 36-39 - In my opinion, it is better to give an example related to plants
L 38-39 - non-coding RNAs – only ncRNAs (the abbreviation appeared in L 35)
L 39 - small ncRNAs (sncRNAs) …
L 40 - Small ncRNAs are – sncRNAs are….
L 43 – sncRNAs - please write abbreviations and then consistently use it
L 49 - Chinese cabbage (Brassica rapa) and poplar (Populus trichocarpa)
L 58 – extracts (add citation).
L 67 – Salicylic acid (SA) is a phytohormone…
L 86-87 - Pseudomonas syringae (P. syringae)
L 87 - WRKY and PR genes, which encode PATHOGENESIS-RELATED PROTEINS such as PR-1, …
L 91-97 - In this study… In my opinion this short fragment should be clarified.
L 99 - The title of this subsection must be clarified. It should be added that changes occur throughout A. thaliana seedlings and in response to hypoxia stress.
L 179-180 - What is the difference between the three pictures (both for SA minus and SA plus) in Figure 2A? I didn't find it anywhere in the description.
L 303-304 – “are likely correlated with the activities of the WRKY network and these factors interact with AtR8 lncRNA and SA in the root.” What the Authors meant when they wrote that WRKY factors interact with SA in the roots?
L 306-307 “ELENA1 lncRNA in Arabidopsis was proposed to be pathogen responsive”. Here it should be added that this has not been fully confirmed in the case of AtR8 lncRNA. Am I right?
340-342- Please describe more precisely on what basis this conclusion drawn.
365-367 - Please re-formulate the conclusion because this is confusing. If I am wrong, please correct it, but in my opinion it would be better to write that SA (20 µM) inhibits AtR8 which promotes root elongation, while SA stimulates WRKY70 and then WRKY53 which together inhibit root elongation.
L 380 - the hypersensitive response (HR) and …
L 471-474 - It would be worth extending the working model and adding both abiotic (hypoxia) and biotic (P. syringae) stress. This would give a full insight into the regulation of the studied processes by AtR8, WRKY and other factors.
Author Response
Thank you very much for your critical review. We carefully consider your comments, the manuscript was revised as below.
Remarks:
Due to the huge amount of results (difficult to interpret), one should write in each subsection of the “Results section” a preliminary conclusion (how the authors interpret the obtained results). The Authors wrote brief conclusions, but in my opinion they should be more complex and extended to facilitate reading of the article.
Answer:
Thank you very much for your suggestion. Indeed, our original brief conclusions in the result section are too simple to understand complex results. To help understanding of readers following brief conclusions are added in end of each result subsections.
Subsection 2.1: L 179-180
Following preliminary conclusion was added in L 179-180:
These results can be inferred that hypoxia induces AtR8 lncRNA participation in the plant defense system including WRKY cascade.
Subsection 2.2: L 204-207
“These results suggest that AtR8 lncRNA, WRKY53, and WRKY70 are all involved in SA signaling at low SA levels, and that AtR8 lncRNA may function in a manner that is antagonistic to the two WRKY proteins.”was changed to:
“The results showed that there was a certain correlation between AtR8 lncRNA and WRKY53 and WRKY70 under the condition of low concentration of SA, and this correlation indicated that AtR8 lncRNA might be antagonistic relationship with these two WRKY proteins in SA-dependent inhibition of primary root elongation.”
Subsection 2.3: L 255-L 258
“Furthermore, mutations in WRKY53 or WRKY70 led to induction of AtR8 lncRNA accumulation, but had no apparent effect on the transcript levels of NPR1 or PR-1, suggesting that PR-1 accumulation was independent of AtR8 accumulation.” was changed to:
“Furthermore, mutations in WRKY53 or WRKY70 led to induction of AtR8 lncRNA accumulation, but had no apparent effect on the transcript levels of NPR1 or PR-1. These results suggesting that PR-1 accumulation was independent of AtR8 accumulation, it means that the AtR8 lncRNA might participate in unknown PR protein-independent defense mechanism.”
Subsection 2.4: L 289-L 293
“These results suggest that WRKY53 or WRKY70 act as repressors in SA-mediated regulation of AtR8 in the short-term presence of SA.” was changed to:
“These results indicate that WRKY53 or WRKY70 act as repressors in SA-mediated regulation of AtR8 in the short-term presence of SA. And the responses of WRKY53 and WRKY70 genes under low level SA are relatively rapid (start within one hour) and fur induced in atr8 mutant.”
Subsection 2.5: L 326-327
Following preliminary conclusion was added in L 326-327:
“These results indicate that the AtR8 lncRNA surely acts on plant defense cooperating with the WRKYs-PR-1 pathway.”
Subsection 2.6: L 350-352
Following preliminary conclusion was added in L 350-352:
“These results suggest that the AtR8 lncRNA functions at early developmental stage after germination, before onset of PR protein syntheses in germinating seeds and tiny seedlings.”
Subsection 2.7: L 374-375
Following preliminary conclusion was added in L 374-375:
“These results suggest that the NPR1 might be a repressor of AtR8 gene as well as an activator of WRKY53 and WRKY70 genes.”
Subsection 2.1 - Based on the results presented in subsection 2.1, can we conclude that AtR8 lncRNA is a repressor of defense responses arising from hypoxia stress in A. thaliana seedlings?
Answer:
Thank you for your question. Looking at microarray results in subsection 2.1 alone, AtR8 lncRNA can be certainly seen as a negative regulator of hypoxia-inducible defense. But we think AtR8 lncRNA functions in germinating seed and tiny seedling before establishment of hypoxia stress relating defense, and supports defense function by another mechanism independent from PR proteins. This hypothesis has been described in the discussion section.
Is there any explanation why hypoxic stress works in contrast to SA in the AtR8 accumulation process?
Answer:
Thank you for your question. We also want to know this point but we don’t have any clear answer for it. Because the AtR8 lncRNA molecule must be multifunctional, defense, root elongation, seed germination and hypoxic stress response, it makes more complicate to understand their molecular mechanism. Plant hypoxic condition particularly occurs in flooding root and main defense also active in root. So, the AtR8, hypoxia and defense all converge in root. Difference of reactivities in AtR8 gene expression between hypoxia and SA might be reflection of AtR8 lncRNA multifunctional aspects in the small spatial area, root. Anyway, we need further studies.
Is there somewhere in the text a discussion of Figure 1C?
Answer:
Sorry, description of Figure 1C was missing in the text, so that following explanation was added in L 180-182,
“In addition, Northern blot analysis using Wt, atr8, wrky53, and wrky70 mutants growing in non-stressed condition showed that AtR8 lncRNA was not detectable in atr8 plants and increasing in wrky53 plants (Fig. 1C).”
L 145-148 - Please consider more than 4 genes for data validation.
Answer:
Data validation of microarray was done by RT-qPCR using 4 genes, WRKY53, WRKY46, WRKY54 and WRKY70, and they showed well consistency. We thought it had been enough for this purpose, and then WRKY53 and WRKY70 genes were used next experiments. While PR-5, PLA2A and PDF1.2 genes were subjected to RT-qPCR, and their up-regulations were confirmed, they were consistent with microarray. These genes are considered for next publication.
Are there any other genes associated with stress/defense reaction (e.g. NPR1 or PR) among differentially expressed genes, which are taken into account in subsequent analyzes?
Answer:
Yes, there are many other defense related genes including AtR8 lncRNA-dependent biological processes, they can be found in our microarray data (Supplemental Table S2). Of course some of them are quite interesting, so we are continuing further studies and considering next publications. Thank you.
Subsection 2.2 - Is there any relationship to root length, SA level and plant response to biotic or abiotic stress? How can explain the inhibition of root elongation in wild-type plants after SA treatment? Additionally, based on the results of section 2.2 one could conclude that AtR8 stimulates root elongation in response to low SA levels, in contrast to the hypoxia-induced defense response?
Answer:
Thank you for your comment. To answer your questions sufficiently, we think we need further substantial experiments, but we try to explain our opinion as much as possible. In general, biotic and abiotic stresses induce SA biogenesis in plant, then diverse SA-dependent responses are caused. To link different phenomena; root elongation and defense, different developmental stages and different organs, we propose ‘growth-defense trade-off’ in Arabidopsis seedlings. Since SA is so-called ‘emergency signal’ for plant, SA induces defense system and pauses growth simultaneously, because available energy resources is limited in small plant. As shown in Figure 7, AtR8 lncRNA is predominantly accumulated in early stage of germinating seeds, then followed by WRKY53 and WRKY70 mRNAs accumulation. Our theory is based on hypothesis that AtR8 lncRNA is involved in additional unknown energy efficient defense. Such discussion is described in section 3.5.
For better understand of SA and root elongation, an additional reference cited (L 81 and L 442).
Subsection 2.3
L 202 “suggesting that PR-1 accumulation was independent of AtR8 accumulation” - please explain because I do not understand the authors' reasoning
Answer:
In Figure 4, the defect of AtR8 lncRNA leads to the rise of the PR1 gene as well as the rise of WRKY53 and WRKY70 (Figure 4C). However, in the WRKY mutants, faint amount of increments in the accumulation of AtR8 lncRNA were observed, but any changes of PR1 were occured. Therefore, we conclude that AtR8 lncRNA and PR1 are independent. Furthermore, Figure 8 using npr1 hypo-mutant plant is also support the conclusion.
Subsection 2.5
In order to draw more complete conclusions, it would be worth examining the endogenous level of salicylic acid in WT plants and individual mutants, and then correlating with the transcriptional activity of the studied genes. The same applies to other analyses, e.g. studies on the impact of hypoxia stress and even those with exogenous application of SA (plants contained a certain level of endogenous SA).
Answer:
Thank you very much for your constructive suggestion. The study of changing endogenous SA level has been considered, so two mutant line of endogenous SA biogenesis have been obtained. Now, we are performing null mutants screening. We hope that research on endogenous SA and other related factors can be carried out in near future.
Minor remarks:
Please correct the citation of papers throughout the manuscript according to the journal guidelines.
Answer:
All citations were corrected.
L 2-4 - In my opinion, the title can be improved because the work contains more results than the title of the work indicates.
Answer:
According to your suggestion, the article title was improved as ‘The Arabidopsis hypoxia inducible AtR8 long non-coding RNA also contributes to plant defense and root elongation coordinating with WRKY genes under low levels of salicylic acid’. (new L 2-5)
L 16-30 - The abstract should be improved. On the one hand, it contains too detailed information (e.g. the sentence "36% of 289 up-regulated genes in atr8, including 12 SA-responsive WRKY genes, were associated with ... "), while on the other hand abbreviations not quite obvious for everyone, such as PR-1 or NPR1.
Answer:
Thank you for your suggestion. The abstract was altered as below.
Abstract: AtR8 lncRNA was previously identified in the flowering plant Arabidopsis as an abundant Pol III-transcribed long non-coding RNA (lncRNA) of ~260 nt. AtR8 lncRNA accumulation is responsive to hypoxic stress and salicylic acid (SA) treatment in roots, but its function has not yet been identified. In this study, microarray analysis of an atr8 mutant and wild-type Arabidopsis indicated a strong association of AtR8 lncRNA with the defense response. 36% of 289 up-regulated genes in atr8, including 12 SA-responsive WRKY genes, were associated with defense or stress responses. AtR8 accumulation exhibited an inverse correlation with accumulation of two WRKY genes (WRKY53/WRKY70) when plants were exposed to exogenous low SA concentrations (20 µM), infected with Pseudomonas syringae, or in the early stage of development. The highest AtR8 accumulation was observed 5 days after germination, at which time no WRKY53 or WRKY70 mRNA was detectable. The presence of low levels of SA resulted in a significant reduction of root length in atr8 seedlings, whereas wrky53 and wrky70 mutants exhibited the opposite phenotype. Taken together, AtR8 lncRNA participates in PATHOGENESIS-RELATED PROTEINS 1 (PR-1)-independent defense and root elongation that are related to the SA response. Mutual regulation of AtR8 lncRNA and WRKY53/WRKY70 is mediated by NONEXPRESSOR OF PATHOGENESIS-RELATED GENE 1 (NPR1). (new L 17-31)
L 16 - AtR8 lncRNA was previously identified in the flowering plant Arabidopsis thaliana as..
Answer:
According to your suggestion, ‘Arabidopsis’ was changed to ‘Arabidopsis thaliana’ (new L 17).
L 21 – WRKY genes
Answer:
According to your previous opinion for improvement of the abstract, this sentence was deleted.
L 36-39 - In my opinion, it is better to give an example related to plants
Answer:
Following description and reference was added in L 39-41:
“For plants, genome-wide search for ncRNAs has been previously performed in a variety of plants such as Arabidopsis thaliana, medicago truncatula, and so on [3].”
L 38-39 - non-coding RNAs – only ncRNAs (the abbreviation appeared in L 35)
Answer:
Thank you for your suggestion, ‘non-coding RNAs’ was deleted. (new L 39)
L 39 - small ncRNAs (sncRNAs) …
Answer:
According to your suggestion, the abbreviation in L 39 was corrected as ‘sncRNAs’. (new L 42)
L 40 - Small ncRNAs are – sncRNAs are….
Answer:
Corrected as your suggestion (new L 42).
L 43 – sncRNAs - please write abbreviations and then consistently use it
Answer:
Corrected as your suggestion (new L 53).
L 49 - Chinese cabbage (Brassica rapa) and poplar (Populus trichocarpa)
Answer:
According to your suggestion, scientific names were added (new L 59-60).
L 58 – extracts (add citation).
Answer:
Cited as 17 (new L 69).
L 67 – Salicylic acid (SA) is a phytohormone…
Answer:
Thank you for your suggestion, we have fixed the abbreviation (new L 78).
L 86-87 - Pseudomonas syringae (P. syringae)
Answer:
Sorry, we are not sure your suggestion in this part (new L 98).
L 87 – WRKY and PR genes, which encode PATHOGENESIS-RELATED PROTEINS such as PR-1, …
Answer:
Thank you for your suggestion, the italic letters were fixed (new L 99).
L 91-97 - In this study… In my opinion this short fragment should be clarified.
Answer:
Thank you very much for your critical review. As you pointed out, this paragraph was not easy to understand. So, this part was revised as below.
In this study, microarray-based transcriptome analysis of an AtR8 lncRNA-defective mutant (atr8) under hypoxic stress revealed a tight interaction between AtR8 lncRNA and defense functions which were induced by low level SA. To further understand the potential defensive role of AtR8 lncRNA, relationships between AtR8 and WRKY53/WRKY70 in the presence of SA or upon infection with P. syringae were also examined. Futhermore, AtR8 lncRNA was also involved in root elongation under low SA concentrations (20 µM). Therefore, this study proposes the importance of AtR8 lncRNA in the post-germination development of early seedlings, which is likely to relate to defense functions in the early developmental stage of plants. (new L 102-109)
L 99 - The title of this subsection must be clarified. It should be added that changes occur throughout A. thaliana seedlings and in response to hypoxia stress.
Answer:
Thank you for your suggestion, “Correlation between transcription of AtR8 lncRNA and WRKY TF genes” have changed to “Correlation between transcription of AtR8 lncRNA and WRKY TF genes in Arabidopsis seedlings under hypoxic stress” (new L 111-112).
L 179-180 - What is the difference between the three pictures (both for SA minus and SA plus) in Figure 2A? I didn't find it anywhere in the description.
Answer:
The three pictures are intended for elimination of media-dependent effects. Wt and every two out of the three mutant plants are combined and seeded in three growing plates. Explanation was added in the legend of Figure 2.
L 303-304 – “are likely correlated with the activities of the WRKY network and these factors interact with AtR8 lncRNA and SA in the root.” What the Authors meant when they wrote that WRKY factors interact with SA in the roots?
Answer:
We performed the same repeated experiments with Arabidopsis roots only, and the results obtained with experiments using whole plants are consistent, and SA inhibits root growth, and AtR8 expression is also in the roots of the plant, so we believe that WRKY network and these factors interact with AtR8 lncRNA and SA in the root. These results are added as new Supplemental Figure S4.
L 306-307 “ELENA1 lncRNA in Arabidopsis was proposed to be pathogen responsive”. Here it should be added that this has not been fully confirmed in the case of AtR8 lncRNA. Am I right?
Answer:
Yes, you are right. We have added in the text as ‘Although this EF-Tu induction has not been confirmed in our study’ (new L 394-395).
L 340-342- Please describe more precisely on what basis this conclusion drawn.
Answer:
Thank you for your suggestion. “Importantly, AtR8 lncRNA accumulation was found to be independent of PR-1 (Fig. 4), and AtR8 lncRNA might therefore contribute to a PR-1-independent defense mechanism.” was changed to;
“Importantly, AtR8 lncRNA accumulation was found to be independent of PR-1, because PR-1 mRNA levels were not changed regardless of AtR8 lncRNA levels in wrky53 and wrky70 (Fig. 4D and 4E), and AtR8 lncRNA might therefore contribute to a PR-1-independent defense mechanism.” (new L 426-429)
L 365-367 - Please re-formulate the conclusion because this is confusing. If I am wrong, please correct it, but in my opinion it would be better to write that SA (20 µM) inhibits AtR8 which promotes root elongation, while SA stimulates WRKY70 and then WRKY53 which together inhibit root elongation.
Answer:
Thank you very much for your suggestion. The conclusion re-formulate as follow (new L 459-462),
“Therefore, we assume that AtR8 lncRNA negatively regulates sensitivity of SA (20 µM) on root elongation, therefore atr8 shows shorter root phenotype comparing with Wt. On the other hand, WRKY53 and WRKY70 positively regulate sensitivity of SA (20 µM), appearing longer root phenotypes in those mutants.”
L 380 - the hypersensitive response (HR) and …
Answer:
Thank you, we have fixed the abbreviation. (new L 466)
L 471-474 - It would be worth extending the working model and adding both abiotic (hypoxia) and biotic (P. syringae) stress. This would give a full insight into the regulation of the studied processes by AtR8, WRKY and other factors.
Answer:
Thank you. According to your suggestion, hypoxia and P. syringae were added in the mode. Hypoxia inhibits AtR8 expression (previously indicated by Wu et al. 2012), and P. syringae infection elevates SA level.